# Auto-Encoding Generative Adversarial Networks towards Mode Collapse Reduction and Feature Representation Enhancement

**DOI:** 10.3390/e25121657

**Published:** 2023-12-13

**Authors:** Yang Zou, Yuxuan Wang, Xiaoxiang Lu

**Affiliations:** Institute of Intelligence Science and Technology, School of Computer and Information, Hohai University, Nanjing 211100, China; yxwang0105@hhu.edu.cn (Y.W.); luxx0824@hhu.edu.cn (X.L.)

**Keywords:** GAN, auto-encoding, cluster center matching, mode collapse, feature representation

## Abstract

Generative Adversarial Nets (GANs) are a kind of transformative deep learning framework that has been frequently applied to a large variety of applications related to the processing of images, video, speech, and text. However, GANs still suffer from drawbacks such as mode collapse and training instability. To address these challenges, this paper proposes an Auto-Encoding GAN, which is composed of a set of generators, a discriminator, an encoder, and a decoder. The set of generators is responsible for learning diverse modes, and the discriminator is used to distinguish between real samples and generated ones. The encoder maps generated and real samples to the embedding space to encode distinguishable features, and the decoder determines from which generator the generated samples come and from which mode the real samples come. They are jointly optimized in training to enhance the feature representation. Moreover, a clustering algorithm is employed to perceive the distribution of real and generated samples, and an algorithm for cluster center matching is accordingly constructed to maintain the consistency of the distribution, thus preventing multiple generators from covering a certain mode. Extensive experiments are conducted on two classes of datasets, and the results visually and quantitatively demonstrate the preferable capability of the proposed model for reducing mode collapse and enhancing feature representation.

## 1. Introduction

Generative Adversarial Nets (GANs) have emerged as a transformative deep learning framework that can closely resemble real-world data [1,2]. Generally, a GAN is composed of a pair of networks, i.e., a generator and a discriminator, which are correlated by an adversarial mechanism. The former tries to produce data that is indistinguishable from real data, while the latter learns to discriminate between real and fake data.

Since their introduction in 2014, GANs, along with their numerous variants, have been frequently exploited in a wide range of applications in many fields related to machine learning, such as image-to-image translation [3], image generation from textual descriptions [4], image style transfer [5], image inpainting [6], image super-resolution [7], video generation [8], data augmentation [9], text transfer [10], time series anomaly detection [11], and many others.

Nevertheless, GANs suffer from several drawbacks, such as training instability, mode collapse, vanishing gradients, the generation of blurred images, etc. [12,13]. Among them, mode collapse and training instability have been crucial challenges during GAN training. Mode collapse indicates a phenomenon in which the generator’s output is constrained to yield repetitive samples that lack the comprehensive range of the real data distribution. In order to remedy these issues, researchers have made a number of improvements, which can be roughly classified into the following categories:Adding constraints: Conditional GANs (CGANs) [14] strengthen the relationship between input and output data using conditional information, forcing the generator to learn diverse modes. The InfoGAN [15] maximizes the mutual information between a subset of the generator’s input and the generated output to obtain distinguishable features so as to avoid mode collapse.Augmenting generators: Typical models include Multi-Agent Diverse Generative Adversarial Networks (MAD-GAN) [16] and Mixture Generative Adversarial Nets (MGAN) [17]. Unlike GANs, their discriminators need not only to distinguish between real samples and generated samples but also to determine from which generator the generated samples come.Modifying loss function: The Wasserstein GAN (WGAN) [18] adopts the Earth-Mover (EM) distance to measure the discrepancy between the generated distribution and the real distribution and minimizes an approximation of the EM distance to make sure the former is as close to the later as possible, to cover multiple modes.Imposing gradient penalty: A variant of the WGAN [19] is proposed to tackle issues originating from the usage of weight clipping to enforce a Lipschitz constraint on the discriminator by penalizing the norm of its gradient with respect to the input.

Note that in the first two categories, the discriminator primarily considers the distribution of generated samples but neglects the distribution of real samples, which could lead to the problem that multiple generators cover different parts of a certain mode, probably resulting in mode collapse. As for the last two categories, even though the generator can discover multiple modes, it still retains the nature of continuous mapping in GANs, which may cover the blank area between modes and generate poor samples, conceivably leading to training instability.

In order to mitigate the drawbacks of mode collapse and training instability in GANs, this paper proposes a variant GAN, called the Auto-Encoding Generative Adversarial Network (AE-GAN), which consists of a set of generators, a discriminator, an encoder, and a decoder. The set of generators accounts for covering diverse modes; the encoder maps generated samples and real samples to the embedding space; and the decoder distinguishes from which generator the generated samples come and from which mode the real samples come. The model is equipped with a cluster center matching algorithm to maintain consistency in the distribution of the real and generated samples. Experimental results on two kinds of datasets show that the AE-GAN can not only reduce mode collapse but also possesses a preferable capability in feature representation.

This paper makes the following technical contributions:A variant GAN, the AE-GAN, that incorporates a set of generators, a discriminator, an encoder, and a decoder is proposed as a result of the qualitative analysis of mode collapse conducted from the perspective of data distribution.An algorithm for cluster center matching is presented to maintain consistency in the distribution of the real and generated samples so as to prevent multiple generators from covering different parts of a certain mode, thus reducing mode collapse.The training algorithm for the proposed model is provided, where the encoder and decoder are jointly optimized using generated samples and real samples, which can reduce the mode collapse of the generators and enhance the feature representation of the encoder.

Extensive experiments are conducted on two classes of datasets, and the results visually and quantitatively demonstrate the preferable capability of the AE-GAN for reducing mode collapse and enhancing feature representation. The code for the AE-GAN is available at the following website: https://github.com/luxiaoxiang0002/Auto-Encoding-GAN/tree/master (accessed on 11 December 2023).

The remainder of this paper is organized as follows: Section 2 reviews the state of the art of related work; Section 3 elaborates on the AE-GAN, including its network architecture, objective function, and training algorithm; Section 4 describes extensive experiments on two categories of datasets; and, finally, Section 5 concludes the paper.

## 2. Related Work

In order to address the issues of training instability and mode collapse in GANs, researchers have introduced quite a few improvements to GANs.

Several variants of GANs have been introduced to tackle training instability. Notably, the instability of the training process can result in mode collapse. Radford et al. [20] introduced a class of models called Deep Convolutional GANs (DCGANs) that impose a bundle of constraints on the architectural topology to stabilize the process of training. Salimans et al. [21] presented a few techniques for training GANs including feature matching and minibatch discrimination. Zhao et al. [22] introduced the Energy-based GAN model (EBGAN) that takes the discriminator as an energy function that assigns low energies to the regions near the data manifold and higher energies to the others. To keep the balance between the discriminator and the generator, Berthelot et al. proposed a Boundary Equilibrium GAN [23] that adopts an equilibrium-enforcing approach paired with a loss derived from the Wasserstein distance to train auto-encoder-based GANs.

Much effort has been made to investigate solutions to the mode collapse problem. Mirza and Osindero [14] introduced a CGAN in which a GAN is extended to be conditioned on auxiliary information, such as class labels or data from other modalities, to enhance the relationship between input data and output data, forcing the generator to learn diverse modes. Metz et al. [24] proposed the Unrolled GAN to stabilize GAN training and reduce mode collapse by defining the generator’s objective with respect to the unrolled optimization of the discriminator, which allows training to be adjusted between using the optimal discriminator in the generator’s objective using the current value of the discriminator. Chen [15] described the InfoGAN, an information-theoretic extension to GANs that maximizes the mutual information between a fixed small subset of a GAN’s noise variables and the observations to avoid the mode collapse of the generator and learn distinguishable and interpretable representations. Eghbal-Zadeh et al. [25] presented a Mixture Density GAN (MD-GAN) that enables the discriminator to create several clusters in its output embedding space for real images and, thus, forces the generator to exploit these and detect different modes in the data.

In VEEGAN [26], a reconstructor network is introduced to reverse the action of the generator by mapping from the data distribution to a Gaussian distribution to discover the modes in it, as the generated data probably coincides with the true data when the reconstructor maps both of them to a Gaussian distribution. A Bayesian framework is employed [27] to optimize GANs with the stochastic gradient Hamiltonian Monte Carlo algorithm, producing interpretable and diverse generated samples. Durugkar et al. [28] proposed a framework called GMAN that extends GANs to include multiple discriminators to avoid mode collapse. MAD-GAN [16] and MGAN [17] are two typical generalized architectures of GANs that incorporate multiple generators and one discriminator. Unlike GANs, the two models are designed not only to discriminate between real and generated samples but also to identify the generator from which the generated sample is derived. Their difference lies in that in the former, two strategies for the modifying generator and the discriminator’s objective functions, respectively, are designed to address the mode collapse issue, whereas, in the latter, an additional classifier is introduced to discriminate samples produced by each generator, which can force generators to generate diverse samples and avoid mode collapse.

WGAN [18] adopts the EM distance to measure the difference between the generated distribution and the real distribution, in terms of theoretical analysis, and minimizes a reasonable and efficient approximation of the EM distance to guarantee the former distribution is as close as possible to the latter, so as to avoid mode collapse. Moreover, it can promote the stability of learning as the EM distance is differentiable. However, sometimes WGAN can still generate poor samples or fail to converge due to the usage of weight clipping to enforce a Lipschitz constraint on the critic. An improved variant of WGAN [19] is accordingly proposed to tackle these issues by penalizing the norm of gradient of the critic with respect to its input.

Our model differs from the above-mentioned multi-generator models mainly in two aspects. First, our model is composed of a set of generators, a discriminator, an encoder, and a decoder, which is architecturally different from the existing models. Moreover, the model is equipped with an algorithm for cluster center matching to maintain consistency between the distribution of the real and generated samples.

## 3. Auto-Encoding GAN

This section first reviews GANs, briefly, and analyzes the issue of mode collapse of GANs from the perspective of data distribution. It then provides the network architecture of the AE-GAN and its objective function in sequence, followed by an algorithm for cluster center matching and the training algorithm of the AE-GAN.

### 3.1. GAN

Generally, a GAN consists of a pair of neural networks, i.e., a generator G and a discriminator D, that are interconnected. The two networks perform in the manner of competing with each other. On the one hand, the network G takes a latent space z from the noise distribution $p_{z}$ as input and produces synthetic samples $G\left(z\right)$ with the objective of generating data that is indistinguishable from real data samples x originating from the probability distribution $p_{data}$. On the other hand, D takes both real samples x from an actual dataset and fake samples $G\left(z\right)$ created by G as the input and determines whether the input data is real or fake.

In the training process, both networks work in a zero-sum game. While G aims to generate more realistic samples, D promotes its capability of distinguishing between real and fake samples. The formal expression of the loss function can be represented as:(1)$$\underset{G}{\min}\underset{D}{\max}L=E_{x\sim p_{data}}\left[\log D\left(x\right)\right]+E_{z\sim p_{z}}\left[\log\left(1-D\left(G\left(z\right)\right)\right)\right]$$
where the probabilities $D\left(x\right)$ and $D\left(G\left(z\right)\right)$ denote the outcomes from D for real and fake samples, respectively. The first term in the equation forces D to correctly classify real data by maximizing $\log D\left(x\right)$, whereas the second term compels G to generate realistic data that D can classify as real by minimizing $\log\left(1-D\left(G\left(z\right)\right)\right)$. The convergence of a GAN can be reached when the generator and the discriminator reach a Nash equilibrium.

### 3.2. Auto-Encoding GAN

The AE-GAN is comprised of a set of generators, a discriminator, an encoder, and a decoder. The set of generators is responsible for covering different modes and, as usual, the discriminator distinguishes between real samples and generated samples, ensuring that the generated distribution does not deviate from the real distribution. In order to prevent multiple generators from covering different parts of a certain mode, a clustering algorithm is introduced to perceive the distribution of real samples and generated samples, and an algorithm for cluster center matching is presented to maintain consistency in the distribution of the real and generated samples. Then, the encoder and decoder are jointly optimized by the generated and real samples. In this manner, the encoder can map generated samples and real samples to the embedding space to encode distinguishable features, and the decoder can distinguish from which generator the generated samples come and from which mode the real samples come.

The architecture of the AE-GAN is depicted in Figure 1.

### 3.3. Adding Generators to Reduce Training Instability and Mode Collapse

From the perspective of data distribution, it has been verified that adding generators is an effective way to tackle training instability and mode collapse.

The training instability of GANs means that it is difficult for the learning processes of the generator G and the discriminator D to converge synchronously. From the perspective of data distribution, a possible reason is that the complexity of real samples’ distributions affects the speed of convergence for G and D. If the real samples are subject to a simple single-mode distribution, G and D will converge quickly and synchronously; otherwise, if the real samples are subject to a complex multi-mode distribution, it will be difficult for G and D to converge synchronously. These two phenomena are illustrated in Figure 2 and Figure 3, respectively.

As shown in Figure 3, it can be seen from the loss curve of the four-mode distribution that the training process of G and D of a GAN is unstable and difficult to converge. On the contrary, in Figure 2, G and D converge quickly and synchronously for a single-mode distribution. Therefore, if one generator is used to learn multiple modes of a distribution, it would be difficult for the GAN to converge and the training process is unstable. Instead, if a set of generators is employed to learn multiple modes, and it is guaranteed that a single generator only covers a single mode, the training instability of the GAN could be effectively settled. An example in this regard is as shown in Figure 4. Furthermore, it can be seen from Figure 3 and Figure 4 that the generators and discriminator of an AE-GAN converge and tend to stabilize much more quickly than those of a GAN.

In essence, mode collapse in GANs refers to the tendency for GANs to concentrate continuously mapped values on a single mode during the training process. The main reason for this phenomenon is that GANs can only approximate continuous mappings. Nevertheless, a multi-mode distribution commonly belongs to the class of discrete distributions that is not continuous. Thus, GANs cannot easily cover discrete multi-mode distributions with continuous mapping. If continuous mapping is forced to cover multiple modes, the values of the continuous mapping would inevitably cover some blank areas outside of the modes. As a consequence, GANs may generate some samples that have no realistic meaning, which explains why GANs can generate poor samples. For theoretical proof of mode collapse in GANs, please refer to [29].

Therefore, the key to addressing mode collapse is to build a discontinuous mapping. Adding generators can discretize the generated distribution such that each generator covers a mode, which is essentially similar to the way that multiple GANs achieve a discontinuous mapping of the multi-mode distribution, but the difference lies in that simply adding generators can do so. Hence, AE-GANs adopt the way of adding generators to realize a discontinuous mapping for the multi-mode distribution, which cannot only cover multiple modes but also reduce the training instability of the model, as shown in Figure 4.

### 3.4. Objective Function of Auto-Encoding GANs

Assuming that real samples are subject to a distribution $P_{data}$ containing k modes, that is, $x\sim P_{data}$, and noise samples are subject to a standard normal distribution, i.e., $z\sim N\left(0,1\right)$. In terms of the architecture of AE-GANs, a set G of k generators $G=\left\{G_{1},G_{2},\cdots ,G_{k}\right\}$ are constructed to attempt to cover those k modes. A discriminator D is responsible for distinguishing between real and generated samples. An encoder (EC) is used to map the generated samples and real samples to the embedding space. A decoder (DC) not only distinguishes from which generator the generated samples come but also distinguishes from which mode the real samples come, where G, D, and encoder-decoder (ED) are deep networks. $y_{G_{i}}$ represents the label corresponding to the samples generated by the ith generator, and $y_{P_{data}}$ represents the label corresponding to the real samples, which is a pseudo label obtained from cluster center matching. The purpose of auto-encoding GAN training is to gain a set of generators G that can cover different modes and an encoder that is capable of representing features. Therefore, the objective function that needs to be optimized for an AE-GAN can be formalized as follows:(2)$$\begin{array}{cc} \underset{G,\mathrm{ED}}{\min}\underset{D}{\max}L\left(G,D,ED\right) & =E_{x\sim P_{data}}\left[\log D\left(x\right)\right]+\sum_{i=1}^{k}E_{z\sim P_{G_{i}}}\left[\log\left(1-D\left(G_{i}\left(z\right)\right)\right)\right]\\ & +\lambda [\sum_{i=1}^{k}y_{G_{i}}E_{z\sim P_{G_{i}}}\left[\log ED\left(G\left(z\right)\right)\right]+y_{P_{data}}E_{x\sim P_{data}}\left[\log ED\left(x\right)\right]] \end{array}$$
where λ is the weight needed to maintain the balance between D and ED.

The objective function is composed of three constituents. The first constituent, the loss of a GAN, is introduced to ensure that the generated data can fit the distribution of real data. The second is the cross-entropy loss of the generated data, which guarantees that the generated data of different clusters can cover different modes on one hand and, on the other hand, the generated data can be utilized as a reference to obtain the clustering labels of the real data, as it is supposed to be classified into the category of the nearest real data after clustering. Accordingly, the third is the cross-entropy loss of the real data, and its category originates from the nearest generated data. From the perspective of the ED loss, it is expected that the model can distinguish between both the real data in different modes and the generated data in different clusters, and when it converges, the generated data in the different clusters can cover different modes of the real data, and the labels of the generated data and real data are consistent.

Theoretically, the convergence of the objective function for AE-GANs is similar to that of GANs. If G, D, and ED have sufficient capacity and training time, the conditions of convergence for them can be given as follows:

**Proposition** **1.***For* G *fixed, the optimal discriminator*  D *is:*
(3)$$D\left(x\right)=\frac{P_{data}}{P_{data}+\sum_{i=1}^{k}P_{G_{i}}}$$

**Proof** **of** **Proposition** **1.**From Equation (2), fixing G and maximizing $L\left(G,D,ED\right)$ can achieve the optimal D. For expressive convenience, let φ be the third term of Equation (2). Since φ is independent of D, it can be viewed as a constant when considering D. Then, the objective function can be expressed as: $$\underset{D}{\max}L\left(G,D,ED\right)=E_{x\sim P_{data}}\left[\log D\left(x\right)\right]+\sum_{i=1}^{k}E_{z\sim P_{G_{i}}}\left[\log\left(1-D\left(G_{i}\left(z\right)\right)\right)\right]+\varphi$$
which can be simplified as follows: $$\begin{array}{c} \underset{D}{\max}L\left(G,D,ED\right)=\int_{x}P_{data}\log D\left(x\right)dx+\int_{z}\sum_{i=1}^{k}P_{G_{i}}\log\left(1-D\left(G_{i}\left(z\right)\right)\right)dz+\varphi \\ =\int_{x}(P_{data}\log D\left(x\right)+\sum_{i=1}^{k}P_{G_{i}}\log\left(1-D\left(D\left(x\right)\right)\right))dx+\varphi \end{array}$$Notably, the maximal value of the integral can be transformed into the maximal value of the integrand, and $G\left(z\right)$ is constantly approaching x. Let $G\left(z\right)\approx x$, based on the Radon–Nikodym theorem, the optimal D—such that the function $f\left(D\left(x\right)\right)=P_{data}\log D\left(x\right)+\sum_{i=1}^{k}P_{G_{i}}\log\left(1-D\left(x\right)\right)$ can reach its maximum in $\left[0,1\right]$—is exactly Equation (3).Obviously, when $P_{data}=\sum_{i=1}^{k}P_{G_{i}}$, D converges to the optimum. □

**Proposition** **2.***For* G *fixed, the optimal encoder-decoder* ED *is:*(4)$$\left\{\begin{array}{c} {ED}_{G}\left(G\left(z\right)\right)=\sum_{i=1}^{k}y_{G_{i}}\\ {\;\;\;ED}_{x}\left(x\right)=y_{P_{data}} \end{array}\right.$$

**Proof** **of** **Proposition** **2.**Note that from Equation (2), fixing G and minimizing $L\left(G,D,ED\right)$ can obtain the optimal ED. For descriptive convenience, let $$\xi =E_{x\sim P_{data}}\left[\log D\left(x\right)\right]+\sum_{i=1}^{k}E_{z\sim P_{G_{i}}}\left[\log\left(1-D\left(G_{i}\left(z\right)\right)\right)\right]$$As ξ is independent of ED, it can be viewed as a constant when considering ED. Then, the objective function can be expressed as:$$\underset{\mathrm{ED}}{\min}L\left(G,D,\mathrm{ED}\right)=\;\xi +\lambda [\sum_{i=1}^{k}y_{G_{i}}E_{z\sim P_{G_{i}}}\left[\log ED\left(G\left(z\right)\right)\right]+y_{P_{data}}E_{x\sim P_{data}}\left[\log ED\left(x\right)\right]]$$For descriptive convenience, ED can be divided into two parts, ${ED}_{G}\left(G\left(z\right)\right)$ and ${ED}_{x}\left(x\right)$, which are responsible for the representation and classification of generated samples and real samples, respectively.
$$\underset{\mathrm{ED}}{\min}L\left(G,D,\mathrm{ED}\right)=\;\xi +\lambda [\sum_{i=1}^{k}y_{G_{i}}E_{z\sim P_{G_{i}}}\left[\log{ED}_{G}\left(G\left(z\right)\right)\right]+y_{P_{data}}E_{x\sim P_{data}}\left[\log{ED}_{x}\left(x\right)\right]]$$Since ED is mainly adopted for classification, according to the nature of the cross-entropy function, the optimal ED that satisfies $\underset{\mathrm{ED}}{\min}\left(G,D,ED\right)$ can be achieved exactly as Equation (4).Obviously, when $P_{data}=\sum_{i=1}^{k}P_{G_{i}}$, D converges to the optimum. At that point, it can be deduced that $G\left(z\right)=x$, that is, $\sum_{i=1}^{k}y_{G_{i}}=y_{P_{data}}$. Consequently, ${ED}_{G}\left(G\left(z\right)\right)={ED}_{x}\left(x\right)=y_{P_{data}}$.Thus, the convergence condition of ED is reached. □

**Proposition** **3.***For the optimal* D *and* ED *fixed, the optimized generator*  G *is:*
(5)$$G=-\log4+2JSD\left(P_{data}∥P_{G}\right)+2\lambda \int_{x}\sum_{i=1}^{k}y_{G_{i}}\log\sum_{i=1}^{k}y_{G_{i}}dz$$
*where* JSD *is the Jensen–Shannon divergence between the two distributions, and* $P_{G}=\sum_{i=1}^{k}P_{G_{i}}$.

**Proof** **of** **Proposition** **3.**Note that from Equation (2), fixing D and ED and minimizing $L\left(G,D,ED\right)$ can achieve the optimal G. Let $P_{G}=\sum_{i=1}^{k}P_{G_{i}}$, and replace the respective terms in Equation (2) using Equations (3) and (4), resulting in the objective function of the form as below:$$\underset{G}{\min}L\left(G,D,ED\right)=E_{x\sim P_{data}}[\log\frac{P_{data}}{P_{data}+P_{G}}]+\sum_{i=1}^{k}E_{z\sim P_{G_{i}}}[\log\left(1-\frac{P_{data}}{P_{data}+P_{G}}\right)]\;+\lambda [\sum_{i=1}^{k}y_{G_{i}}E_{z\sim P_{G_{i}}}[\log\sum_{i=1}^{k}y_{G_{i}}]+y_{P_{data}}E_{x\sim P_{data}}[\log y_{P_{data}}]]\;=2\log\frac{1}{2}+\int_{x}P_{data}\log\frac{P_{data}}{\left(P_{data}+P_{G}\right)/2}dx+\int_{x}P_{G}\log\frac{P_{G}}{\left(P_{data}+P_{G}\right)/2}dx\;+\lambda [\int_{z}\sum_{i=1}^{k}y_{G_{i}}[\log\sum_{i=1}^{k}y_{G_{i}}]+y_{P_{data}}[\log y_{P_{data}}]dz]$$When $P_{data}=\sum_{i=1}^{k}P_{G_{i}}$, it can be deduced that $G\left(z\right)=x$, that is, $\sum_{i=1}^{k}y_{G_{i}}=y_{P_{data}}$. Therefore, ${ED}_{G}\left(G\left(z\right)\right)=y_{P_{data}}={ED}_{x}\left(x\right)$. Hence, the convergence condition of ED is reached. The equation can be further simplified as:$$\underset{G}{\min}L\left(G,D,ED\right)=-\log4+KL(P_{data}∥\frac{P_{data}+P_{G}}{2})+KL(P_{G}∥\frac{P_{data}+P_{G}}{2})\;+2\lambda \int_{z}\sum_{i=1}^{k}y_{G_{i}}[\log\sum_{i=1}^{k}y_{G_{i}}]dz\;=-\log4+2JSD\left(P_{data}∥P_{G}\right)+2\lambda \int_{z}\sum_{i=1}^{k}y_{G_{i}}[\log\sum_{i=1}^{k}y_{G_{i}}]dz$$
where KL is the Kullback–Leibler divergence and JSD is the Jensen–Shannon divergence.When $P_{data}=P_{G}$, $JSD\left(P_{data}∥P_{G}\right)=0$. As both $y_{G_{i}}$ and $y_{P_{data}}$ are known, the generator converges at that point $G=-\log4+2\lambda \int_{z}\sum_{i=1}^{k}y_{G_{i}}\log\sum_{i=1}^{k}y_{G_{i}}dz$. □

The following conclusion directly follows from Propositions 1–3.

**Theorem** **1.***The objective function of an AE-GAN formalized in Equation (2) converges*.

### 3.5. Cluster Center Matching

If the decoder only distinguishes from which generator the generated samples come, it may cause the generated samples to cover different parts of a certain mode, resulting in mode collapse. Here, the training results of the InfoGAN and the MGAN using a 2D dataset are taken as an example to showcase the situations of mode collapse, as depicted in Figure 5.

If the mode distribution of real samples is taken into account, the classifier can effectively avoid mode overlap. Nevertheless, the mode distribution of real samples is unknown beforehand, so how to approximate the distribution of real samples becomes a key issue. Both the generated distribution and real distribution indicate the distributions of the embedding space output from the encoder. Since the same mode is frequently clustered in the feature space due to the similarity between real samples, a clustering algorithm can be employed to perceive the distribution of the mode of the real samples.

In order to prevent the perceived result from deviating from the real distribution, we took the generated distribution as a reference and reduced the deviation by minimizing the distances between the cluster centers of the generated distribution and the real distribution. The fundamental reason why the generated distribution can be used as a reference is that it will be constantly approaching the real distribution during the training process. Theoretically, when a trained model converges, the generated distribution can approximate the real distribution; however, in practical training, the encoder is not able to accurately learn all the characteristics of a real distribution due to the fact that the generated distribution cannot completely reflect the real distribution. If some predicted information of the real distribution can be supplemented into the training process, the learning capability of the encoder for the real distribution will be improved accordingly.

When the generated samples and the real samples are fed into the encoder, the encoder will produce corresponding embedded features, denoted by $h_{x}=EC\left(x\right)$ and $h_{G\left(z\right)}=EC\left(G\left(z\right)\right)$, respectively. By applying a clustering algorithm to the real samples and generated samples, their cluster centers can be obtained, denoted as $\mu_{x}=\left\{\mu_{x_{1}},\mu_{x_{2}},\cdots ,\mu_{x_{k}}\right\}$ and $\mu_{G\left(z\right)}=\left\{\mu_{G_{1}\left(z\right)},\mu_{G_{2}\left(z\right)},\cdots ,\mu_{G_{k}\left(z\right)}\right\}$, respectively. In order to match the cluster centers of the real distribution and the generated distribution, that is, to minimize the distances between the cluster centers of the real distribution and the generated distribution, the loss function that needs to be satisfied is formalized as follows:(6)$$\min L_{c}\left(G,\;EC\right)=\sum_{i=1}^{k}\left(E_{x\sim P_{data}}{\left\|h_{x}-\mu_{x_{i}}\right\|}^{2}+E_{z\sim P_{G_{i}}}{\left\|h_{G_{i}\left(z\right)}-\mu_{G_{i}\left(z\right)}\right\|}^{2}+{\left\|\mu_{x_{i}}-\mu_{G_{i}\left(z\right)}\right\|}^{2}\right)$$

Combined with the loss function, the process of an algorithm for cluster center matching, with three main steps, is elaborated on as follows:

First, the encoder takes the generated samples and the real samples as the input and outputs the respective embedded features. Then, the k-means++ algorithm is utilized to cluster the generated samples and the real samples in the embedded space, obtaining the clustering center sets $\mu_{G\left(z\right)}$ and $\mu_{x}$ of the generated samples and the real samples, respectively.Second, to minimize the distances between the cluster centers of the generated samples and the real samples, a distance matrix $M_{k\times k}$ of the center sets $\mu_{x}$ and $\mu_{G\left(z\right)}$ is calculated. In terms of $M_{k\times k}$, for each cluster center $\mu_{x_{i}}$, the closest center $\mu_{G_{i}\left(z\right)}$ is matched such that the cumulative sum of ${\left\|\mu_{x_{i}}-\mu_{G_{i}\left(z\right)}\right\|}^{2}$ is the smallest.Third, to unify the cluster assignment of the real samples and the generated samples, the generated samples are used as a reference to keep the matched center pairs $\left(\mu_{x_{i}},\;\mu_{G_{i}\left(z\right)}\right)$ consistent with the corresponding cluster labels.

Through the above process, the loss of cluster center matching between the real distribution and the generated distribution can be obtained, and the generators and encoder can be further optimized using the loss, accordingly.

### 3.6. Training of the Auto-Encoding GAN

The algorithm for training the Auto-Encoding GAN is shown in Algorithm 1.

The algorithm is mainly composed of four parts. The first part conducts the updating of the discriminator D. In the second part, the k-means++ algorithm is first adopted to gain the cluster centers μx and μGz of the real samples x and the generated samples Gz, respectively; then, the algorithm presented in Section 3.5 is harnessed to match the cluster centers between μx and μGz. Next, the matched results are utilized to guide the assignment of the cluster labels. After that, the last two parts sequentially perform the updating of the encoder-decoder ED and the generators G.
**Algorithm 1** Training of the Auto-Encoding GAN**input:** Dataset: X, number of generators: k, training epochs: n;**output:** Generators: G, discriminator: D, encoder-decoder: ED;1: Initialize generators G, discriminator D, and encoder-decoder ED;2: **for** epoch=1 to n **do**3:   Sample a batch x from X;4:   Sample z~N0, 1;5:   Calculate LD=Ex~Pdatalog⁡Dx+∑i=1kEz~PGilog⁡1−DGiz, according to Equation (1);6:   Update D by ascending along the gradient ∇DLD;7:   Cluster hx,hGz using the k-means++ algorithm to obtain cluster centers μx, μGz;8:   Match cluster centers between μx and μGz using the algorithm introduced in Section 3.5;9:   Assign cluster labels yPdata,yG for hx,hGz according to matched cluster centers;10:    Calculate LED=∑i=1kyGiEz~PGilog⁡EDGz+yPdataEx~Pdatalog⁡EDx;11:    Update ED by descending along the gradient ∇EDLED;12:    **for** i=1 to k **do**13:     Calculate LGi=∑i=1kEz~PGilog⁡1−DGiz+λ∑i=1kyGiEz~PGilog⁡EDGiz;14:     Update Gi by descending along the gradient ∇GiLGi;15:    **end for**16:  **end for**

## 4. Experiments and Analysis

In this section, we conduct experiments on both two-dimensional synthetic datasets and image datasets to visually and quantitatively demonstrate the effectiveness of the AE-GAN for reducing mode collapse and enhancing feature representation, respectively. In addition, we conduct experiments on the selection of the hyperparameter indicating the number of generators in the proposed model.

### 4.1. Implementation

To facilitate the reproduction of the experiments, we provide necessary experimental details. Prior to cluster center matching, the k-means++ algorithm is used to attain the cluster centers of the generated and real samples. In each iteration, the algorithm adopts the default parameters of the Python package “sklearn”, except for the number of clusters that need to be specified.

It can be seen from the loss function that λ is a crucial hyperparameter, which is introduced to maintain the balance between the discriminator and the encoder-decoder; their losses can be unified to a magnitude according to practical values in the training. In our experiments, λ is set to 0.5. Moreover, another crucial hyperparameter k determines the number of generators in the AE-GAN, and it is generally set to be the number of categories in the dataset involved in the experiments.

### 4.2. Experiments on Synthetic Datasets

**Synthetic Datasets.** The 2D datasets selected for the experiments are Aggregation, R15, and S1, which are commonly used for clustering. Each cluster in these datasets represents a mode. Aggregation contains 788 samples belonging to five clusters, and the sizes of clusters are different. R15 involves 600 samples of 15 clusters, and each cluster conforms to the Gaussian distribution, but the intervals between them are diverse. S1 includes 5000 samples within 15 clusters, and the shapes of the clusters are different.

**Network Structure.** The generators and the discriminator of all the involved models adopt uniform network structures. The encoder has the same convolutional layers as the discriminator. The decoder is established as a fully connected layer, and the activation function for the classification is *softmax*. All the generators of the AE-GAN adopt the same network structure. The concrete network structures of the models for the synthetic datasets are shown in Table A1 of Appendix A. The other parameters are uniformly set for all models as follows: Adam is used as the optimizer, where the learning rate is set to 0.0004, betas are (0.5, 0.99), the batch size is 128, and the number of epochs is 500. 

**Experimental Results.** Generally, four baseline models, a WGAN, DCGAN, InfoGAN, and MGAN, are selected for comparison in terms of two criteria. First, they are representative of a certain category of GAN variants, and, second, they are more structurally relevant to our model. As a WGAN and DCGAN cannot represent multi-mode, only InfoGAN and MGAN are chosen for comparison here. The objectives of the experiments are to analyze the capabilities of the mode coverage and the feature representation of the models.

The degree of mode coverage of the generator is primarily illustrated using visible results, and the feature representation ability of the models is mainly displayed using the classification results.

**(1)** 
**Experiments for mode coverage**


The experimental results are illustrated in Figure 6, where the degree of mode coverage is visually displayed.

Clearly, although both the InfoGAN and the MGAN can cover most of the given modes, they are prone to mode overlapping. In addition, a proportion of generated samples appear in the blank areas between the modes, especially in the diagrams for Aggregation and S1. In contrast, the AE-GAN can not only properly cover all the modes on all the datasets but also concentrate on the modes, that is, the generated samples do not appear in the blank areas between the modes. Comparatively, the mode coverage performed by the AE-GAN is more comprehensive and reasonable.

**(2)** 
**Experiments for feature representation**


The experimental results are illustrated in Figure 7, where the effects of feature representation for these models are visually displayed using the classification results.

Noticeably, the clusters for each dataset are the same as in Figure 6, represented by the clustered blue points, and the generated samples that are classified by a model as a cluster are indicated by a uniform number with a specific color. Consequently, the accuracy of the classification results of a model can be visually exhibited as the purity of coloring for the clusters. 

Apparently, InfoGAN and MGAN are able to only cover most rather than all modes, as shown in the diagrams for Aggregation and S1, where two or more clusters can be classified as one mode. Moreover, these two models are not capable of correctly classifying the missing modes, as shown for Aggregation, R15, and S1, where one cluster can be recognized as multiple modes. Consequently, an unreasonable division of the modes would result in terms of classification. In contrast, the AE-GAN can cover all the modes for these datasets and accomplishes preferable mode division through a more accurate and reasonable classification.

### 4.3. Experiments on Image Datasets

**Image Datasets.** We choose frequently-used datasets, including USPS, MNIST, Fashion-MNIST, Coil-20, and Cifar-10. USPS and MNIST [30] are typical digital image datasets. USPS contains 9298 images (16 × 16) in 10 categories, where 7291 images are used for training and 2007 images for testing. MNIST contains 70,000 images (28 × 28) in 10 categories, including 60,000 training images and 10,000 testing images. Fashion-MNIST is similar to MNIST, except that it contains different categories [31]. Coil-20 [32] contains 1440 images (128 × 128) falling into 20 categories. Cifar-10 [33] involves 60,000 color images (32 × 32) in 10 categories, including 50,000 for training and 10,000 for testing.

**Evaluation Indicators.** The generator can be evaluated by the quality and diversity of the generated images, and the Frechet Inception Distance (FID) [34] is frequently used for this purpose. The lower the FID score, the closer the generated distribution is to the real distribution, which means that the quality of the generated images is higher and the diversity is better.

Regarding the evaluation of the encoder-decoder, the encoder’s capability for feature representation can be verified from the classification of the images. Clustering Accuracy (ACC) and Normalized Mutual Information (NMI) are usually adopted as evaluation indicators for feature representation, especially for unsupervised learning. The larger the value of ACC or NMI, the better the capability for feature representation.

**Network Structure.** The network structure of a DCGAN [20] is adopted by the involved models for the image datasets USPS, MNIST, and Fashion-MNIST. The network structures for the Coil-20 and Cifar-10 datasets are listed in Table A2 of Appendix A. The other parameters are uniformly set, as previously described.

**Experimental Results.** As stated previously, the WGAN, DCGAN, InfoGAN, and MGAN are selected for comparison. The experiments are conducted on five image datasets to investigate the capabilities of these models with respect to two aspects—the degree of mode coverage and the effect of feature representation. In the image datasets, each category of images is regarded as a mode. 

**(1)** 
**Experiments for mode coverage**


The degree of model coverage is evaluated using the FID. The corresponding experimental results are shown in Table 1.

It is obvious from Table 1 that our AE-GAN model is better than all the baseline models for the USPS, MNIST, Fashion-MNIST, and Coil-20 datasets. Even for the Cifar-10 dataset, the AE-GAN outperforms all the other models, except for InfoGAN, whose FID is 4.64 (5.43%) lower than that of our model. 

As for the average FID across all the datasets, that of the AE-GAN is 29.85 (38%) lower than that of the WGAN, 10.78 (18.60%) lower than that of the DCGAN, 3.21 (6.37%) lower than that of the InfoGAN, and 5.14 (9.83%) lower than that of the MGAN, which demonstrates that the overall performance of the AE-GAN is significantly better than that of the other models in reducing mode collapse.

**(2)** 
**Experiments for feature representation**


As both the WGAN and DCGAN are not able to represent multi-mode, only InfoGAN and MGAN are chosen for the comparative experiments. The metrics NMI and ACC are adopted to evaluate the models and the experimental results are shown in Table 2 and Table 3, respectively.

It is obvious from Table 2 and Table 3 that our AE-GAN model outperforms the other models for the USPS, MNIST, and Coil-20 datasets in terms of the two metrics. For the Fashion-MNIST dataset, the NMI value of the AE-GAN is higher than that of the InfoGAN, whereas its ACC value is lower than that of the InfoGAN. For the Cifar-10 dataset, the InfoGAN performs slightly better than the AE-GAN based on both metrics. A possible reason for this phenomenon is that, compared with multi-generator models, the InfoGAN can randomly choose the hidden code to prevent the model from overfitting.

As for the average metric values across all the datasets, the NMI of the AE-GAN is 1.49 (2.38%) and 4.22 (7.06%) higher than that of the InfoGAN and the MGAN, and the ACC is 2.18 (3.43%) and 3.43 (5.50%) higher than that of the InfoGAN and the MGAN, respectively, which states that the AE-GAN manifests an overall preferable capability for feature representation among the involved models.

### 4.4. Experiments on the Selection of the Number of Generators

In this subsection, we conduct a set of experiments on the synthetic dataset R15 to address the influence of the hyperparameter k that indicates the number of generators on the effectiveness of the AE-GAN. Since R15 contains 15 clusters, we choose five different values of k around the number 15. The experimental results are shown in Figure 8.

It is evident from Figure 8 that, as the hyperparameter controlling the number of generators, the value of k actually affects the overall performance of the proposed model. 

If the value of k equals the number of modes in the real data, the AE-GAN can precisely cover all the modes and achieve accurate feature classification, as shown in Figure 6 and Figure 7, respectively. 

When k is set to a value less than the number of modes, one generator may attempt to cover two or even more modes, and two or more clusters may be simultaneously labeled by one class number. Specific examples are the classes labeled 0, 3, and 4 in the case of k=12, and the class labeled 7 in the case of k=14, as shown in Figure 8. Nevertheless, in these situations, the model can converge as usual and gain acceptable outcomes.

Conversely, when k is larger than the number of modes, one mode may be covered by two or more generators, and one cluster may be labeled by two or more class numbers. Specifically, two generators (corresponding to the classes 8 and 15) try to cover the central mode in the case of k=16, and the two classes 1 and 9, 8 and 10, 4 and 15, and 6 and 15 attempt to occupy a mode, respectively, in the case of k=18. When k=20, class 18 competes with class 15 to cover a mode, and classes 5, 10, 13, and 8 are compelled to cover blank areas between the modes. However, it can be clearly observed from Figure 8 that even in these situations, the model can stably converge and accomplish satisfactory mode division via accurate feature classification.

Without doubt, the above two situations become more and more frequent when k gets further away from the actual number of modes in the real data. Consequently, the condition for applying the model is to pursue a proper value of k that approximates the actual number of modes, which is similar to the issue of determining the number of classes prior to the usage of the well-known k-means algorithms. Accordingly, some techniques, such as the elbow method and the silhouette coefficient, could be employed for this purpose. 

Furthermore, note that the AE-GAN is still superior to the InfoGAN and the MGAN, even in some situations where k differs from the actual number of modes. For instance, in the case k=14, shown in Figure 8, each generator of the AE-GAN exactly covers a mode, except the one corresponding to class 7, which tries to cover two modes. In contrast, three modes are not completely covered by the samples labeled classes 14, 1, and 3 by the InfoGAN, and the situation is even worse for the MGAN, where two modes are almost not covered, as depicted in Figure 6. Therefore, the AE-GAN undoubtedly outperforms the MGAN in reducing mode collapse. On the other hand, most of the modes are accurately divided by the AE-GAN except these two, which are classified as one mode, labeled 7, as shown in Figure 8. Meanwhile, the situation in which one cluster is classified as two modes occurs twice and once in the InfoGAN and the MGAN, respectively, as illustrated in Figure 7. Thus, the AE-GAN evidently outperforms the InfoGAN in feature classification. Hence, as for the overall performance, the AE-GAN with k=14 is preferable to these two models. 

Apparently, similar situations occur in the cases of k=16 and k=18 due to the fact that the AE-GAN achieves much better outcomes in the feature classification. To sum up, when k is extended from the actual value 15 to the other numbers in the range of [14,18], the AE-GAN can still achieve preferable performance. Therefore, there exists a feasible range of *k* around the actual number of modes in the datasets.

## 5. Conclusions

This paper proposed a variant GAN, the AE-GAN, to reduce the mode collapse and enhance the feature representation of GANs. It is composed of a set of generators, a discriminator, an encoder, and a decoder. The generators account for learning diverse modes so as to prevent mode collapse; the discriminator distinguishes between real samples and generated samples; the encoder maps the generated samples and the real samples to the embedding space, encoding distinguishable features among modes; and the decoder decides from which generator the generated samples come and from which mode the real samples come. In order to promote the feature representation of the encoder and prevent multiple generators from covering a certain mode, a clustering algorithm is utilized to perceive the distribution of the real and the generated samples, and an algorithm for cluster center matching is presented to maintain consistency in the distribution of the real and the generated samples. Moreover, the encoder and the decoder are jointly optimized using the generated and the real samples. Experiments were conducted on 2D and image datasets, respectively, to visually and quantitatively demonstrate the effectiveness of the AE-GAN in reducing mode collapse and enhancing feature representation.

In future work, we shall explore other ways of approximating the actual value of the hyperparameter controlling the number of generators in the AE-GAN from prior knowledge of the given data, and we will attempt to enlarge the application scope of the model by leveraging other algorithms to cluster the generated and the real samples as the k-means algorithms are only applicable to a limited number of data distributions. 

## Figures and Tables

**Figure 1 entropy-25-01657-f001:**
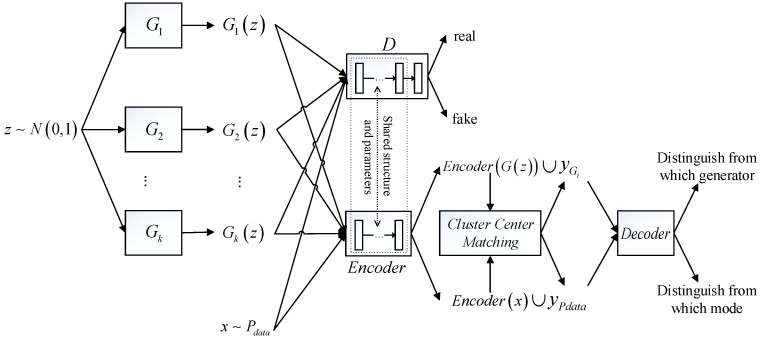
The architecture of the AE-GAN. First, $z\sim N\left(0,1\right)$ is input into multiple generators $G_{1},G_{2},\cdots ,G_{k}$. Then, the generated samples $G_{i}\left(z\right)$ from $G_{i}$ (1≤i≤k) and real samples $x\sim P_{data}$ are simultaneously input into the discriminator and encoder-decoder, respectively. The discriminator, except for the output layer, shares the network structure and parameters with the encoder. The discriminator distinguishes between real and fake samples, whereas the encoder extracts their features. The pseudo labels $y_{P_{data}}$ and $y_{G_{i}}$ of the real and generated samples are, respectively, determined by the cluster center matching algorithm. Finally, taking the outcomes of the encoder along with the pseudo labels as input, the decoder distinguishes from which generator the generated samples come and from which mode the real samples come using cross-entropy loss.

**Figure 2 entropy-25-01657-f002:**
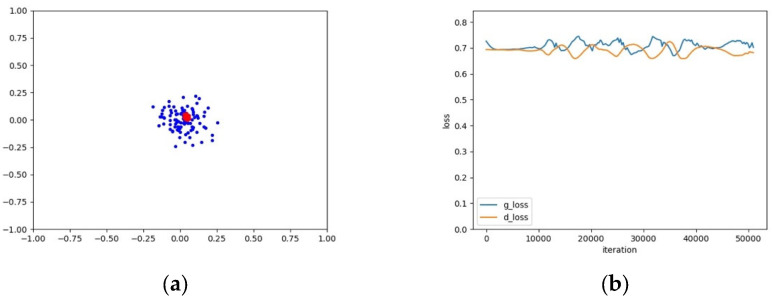
An example of the mode coverage and loss curve of a GAN for a simple single-mode distribution. (**a**) The mode coverage of the GAN, where blue and red points represent real and generated samples, respectively; (**b**) The loss curve of G and D of the GAN.

**Figure 3 entropy-25-01657-f003:**
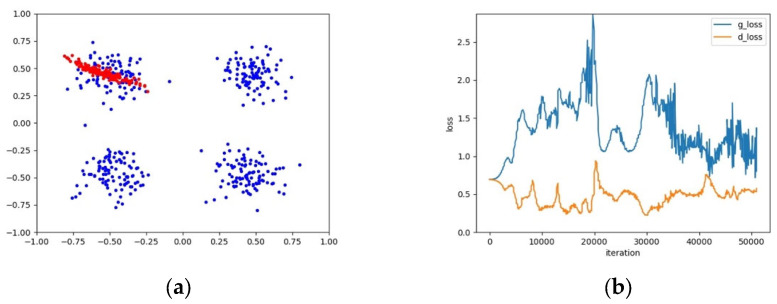
An example of the mode coverage and loss curve of a GAN for a complex four-mode distribution. (**a**) The mode coverage of the GAN, where blue and red points represent real and generated samples, respectively; (**b**) The loss curve of G and D of the GAN.

**Figure 4 entropy-25-01657-f004:**
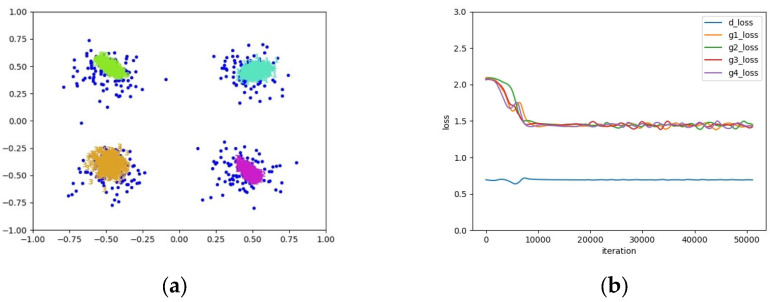
An example of the mode coverage and loss curve of an AE-GAN on a complex four-mode distribution. (**a**) The mode coverage of the AE-GAN, where blue points represent real samples and the others represent generated samples in which samples generated from different generators are denoted by different colors; (**b**) The loss curve of G and D of the AE-GAN.

**Figure 5 entropy-25-01657-f005:**
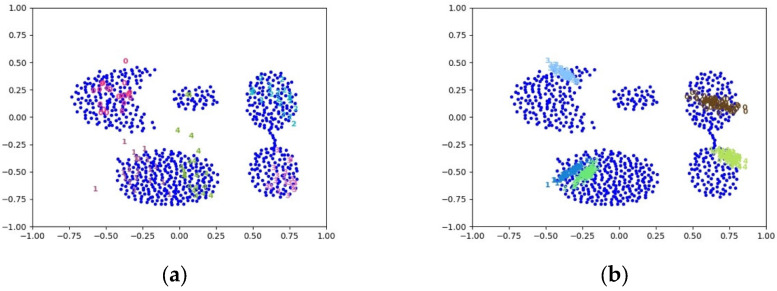
An example of the mode collapse of two models using a 2D dataset, where blue points represent real samples and numbers with other colors represent generated samples. (**a**) The result of the InfoGAN; (**b**) The result of the MGAN.

**Figure 6 entropy-25-01657-f006:**
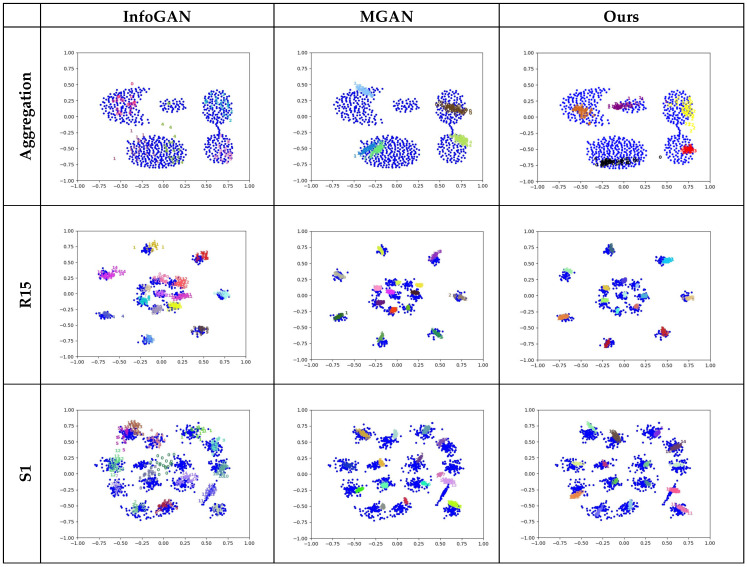
Mode coverage of the involved models using three datasets. Each row exhibits experimental results of all the models for a certain dataset and each column depicts experimental results of a specified model for all the datasets, where blue points represent real samples and numbers with other colors represent the generated samples in the different clusters.

**Figure 7 entropy-25-01657-f007:**
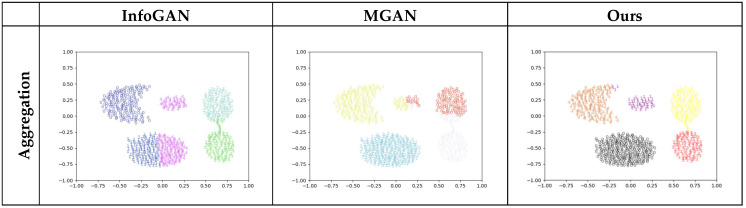
Feature classification of the involved models using three datasets. Each row exhibits experimental results of all the models for a certain dataset and each column depicts experimental results of a specified model for all the datasets, where each number represents a generated sample and simultaneously indicates the cluster to which the generated sample belongs.

**Figure 8 entropy-25-01657-f008:**
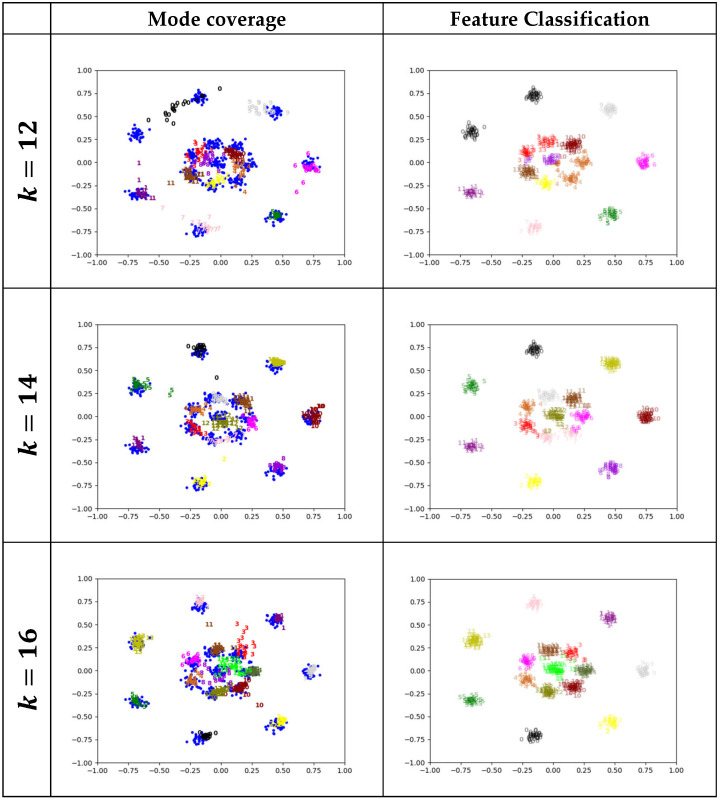
Mode coverage and feature classification of the AE-GAN with different numbers of generators for the dataset R15. Each row depicts the results of the mode coverage and feature classification with respect to a certain number of generators.

**Table 1 entropy-25-01657-t001:** FID values of all the involved models for five image datasets.

Model	USPS	MNIST	Fashion-MNIST	Coil-20	Cifar-10
WGAN	60.74	83.86	82.79	57.46	100.25
DCGAN	51.41	28.70	72.78	41.39	95.47
InfoGAN	34.06	26.71	72.92	37.41	**80.82**
MGAN	30.85	25.29	81.24	36.92	87.23
**Ours**	**28.42**	**22.07**	**64.63**	**35.27**	85.46

The best value in each column is in bold and the second best is underlined.

**Table 2 entropy-25-01657-t002:** NMI (%) of the involved models for five image datasets.

Model	USPS	MNIST	Fashion-MNIST	Coil-20	Cifar-10
InfoGAN	72.57	85.27	59.49	74.70	**20.49**
MGAN	75.99	85.01	54.11	66.49	17.26
**Ours**	**76.21**	**87.24**	**59.83**	**79.34**	18.34

The best value in each column is in bold and the second best is underlined.

**Table 3 entropy-25-01657-t003:** ACC (%) of the involved models for five image datasets.

Model	USPS	MNIST	Fashion-MNIST	Coil-20	Cifar-10
InfoGAN	71.42	92.64	**58.44**	60.41	**34.80**
MGAN	74.13	92.52	56.09	56.53	32.17
**Ours**	**74.92**	**93.27**	58.11	**68.94**	33.34

The best value in each column is in bold and the second best is underlined.

## Data Availability

The data presented in this study are available on request from the corresponding author.

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
