# Peer review of "Auto-Encoding Generative Adversarial Networks towards Mode Collapse Reduction and Feature Representation Enhancement"

_entropy, 2023, doi:10.3390/e25121657_

Round 1

Reviewer 1 Report

Comments and Suggestions for Authors

Overall an interesting approach towards dealing with the mode collapse issue of adversarial networks. Especially interesting is the explicit inclusion of clustering to drive mode disambiguation. The paper appears to have adequate theoretical and experimental justication. It presents experimental evaluation results using both synthetic and real datasets.

In my opinion the paper could benefit from the following:

(1) An extended discussion, conclusions as well as pointers to future work.

(2) It could perhaps provide additional information regarding the inclusion of clustering, e.g. how does it affect the efficiency of the training process in realistic conditions, whether and when could it introduce completitive conditions during training, etc.

(3) Better and more accurate description of the evaluation, including implementation decisions, optimiser decisions, etc.

(3.1) Ideally, and to aid reproducibility, links to code repositories and additional information would be extremely useful.

(4) Certain figures could be provided in higher resolution, e.g. fig 7. 

Comments on the Quality of English Language

I have identified certain minor points that could be improved, eg. on line 205 "it verifies" should read "it has been verified". The authors should double check. 

More importantly, the authors should double check that the citation numbers correspond to the right piece of literature. E.g. on line 472, [34] should be [35].

Author Response

Reviewer 1:

Overall an interesting approach towards dealing with the mode collapse issue of adversarial networks. Especially interesting is the explicit inclusion of clustering to drive mode disambiguation. The paper appears to have adequate theoretical and experimental justification. It presents experimental evaluation results using both synthetic and real datasets.

Answer:

We are grateful to the reviewer for his/her constructive and valuable comments and suggestions that help us considerably improve the quality of the manuscript. The modified places in the revised manuscript are highlighted in blue.

We thank the reviewer for acknowledging our contributions.

In my opinion the paper could benefit from the following:

  • An extended discussion, conclusions as well as pointers to future work.

Answer:

As suggested, we added a separate section, Section 4.4, to conduct more discussion on the proposed method, and added future work to the “Conclusion” section.

Please refer to Section 4.4 on Pages 15-16, and Section 5 on Page 16. The modified contents are highlighted in blue.

(2) It could perhaps provide additional information regarding the inclusion of clustering, e.g., how does it affect the efficiency of the training process in realistic conditions, whether and when could it introduce competitive conditions during training, etc.

Answer:

Clustering is conducted in the feature layer using k-means++ algorithms. The time complexity of it is O(n), where n is the number of samples. Thus, the clustering process is very efficient, and it doesn’t affect the efficiency of training, which can be verified by the process of training. Besides, it will not introduce any competitive conditions during training.

(3) Better and more accurate description of the evaluation, including implementation decisions, optimizer decisions, etc.

Answer:

We added a section “Appendix A” and modified the corresponding paragraphs to supplement concrete network structures and for the involved models and detailed settings of the optimizer. Please refer to Lines 418-425 on Page 12, Lines 490-493 on Page 14, and Appendix A on Page 17.

(3.1) Ideally, and to aid reproducibility, links to code repositories and additional information would be extremely useful.

Answer:

We uploaded the code of the proposed model along with explanatory text to GitHub, and the website is listed below:

https://github.com/luxiaoxiang0002/Auto-Encoding-GAN/tree/master.

Please refer to Lines 90-93 on Page 2.

(4) Certain figures could be provided in higher resolution, e.g., fig 7. 

I have identified certain minor points that could be improved, e.g., on line 205 "it verifies" should read "it has been verified". The authors should double check. 

More importantly, the authors should double check that the citation numbers correspond to the right piece of literature. E.g., on line 472, [34] should be [35].

Answer:

We updated Figures 6 and 7 with pictures in higher resolution. Please refer to Pages 12-13.

We corrected the sentence on Line 205 (now Line 203) on Page 5, and double checked the text thoroughly.

We made corrections to Line 472 (now Line 481) on Page 14, and double checked all the citations in text and made sure that each citation corresponds to the right paper in the references.

Reviewer 2 Report

Comments and Suggestions for Authors

This paper proposes a new network to reduce the mode collapse in GAN by the multiple generators with AE scheme to distinguish from which generate and which mode. Overall methodology is reasonable and novel. However, there are some theoretical and practical limitations. The detail comments are as follows.

1. In this proposed method, the number of generators, k should be matched to the number of actual modes to be efficiently worked. That means we should know the number of modes in advance, and if the number of modes is a lot, then too many generators should be designed. Thus, authors should verify the proposed network to be efficient in the limited number of k, even though the number of modes is much greater than k. I believe the mode collapse might not be reduced in this case. Please add the corresponding experiments to verify the efficiency of the proposed method.

2. Regarding the overall cost function with the constraint for the encoder and the decoder scheme, it cannot be understood that regulating parameter lambda has a minus value. Please explain why the augmented cost function is added with minus lambda regulated constraint. Also, it needs to explain the value of lambda was selected as 0.5 in experiments.

3. Regarding the figure1, this figure does not show the relation between ED scheme and discriminator. That is, overall cost function includes the constraint for ED to be optimized. However, the figure1 does not show any connection between these. Please correct the figure to show this relation in proper way.

4. I agree it can be reasonable approach that the ED scheme is added to match the mode and the corresponding generator by clustering the data and the sample outputs of generators and using each means. However, it is not clear why the constraint for ED is added to the original cost function of GAN scheme. This should be explained theoretically in detail.

Author Response

Reviewer 2:

This paper proposes a new network to reduce the mode collapse in GAN by the multiple generators with AE scheme to distinguish from which generate and which mode. Overall methodology is reasonable and novel.

Answer:

We are grateful to the reviewer for his/her constructive and insightful comments and suggestions that help us considerably improve the quality of the manuscript. The modified places in the revised manuscript are highlighted in blue.

We thank the reviewer for acknowledging our contributions.

However, there are some theoretical and practical limitations. The detail comments are as follows.

  1. In this proposed method, the number of generators, k should be matched to the number of actual modes to be efficiently worked. That means we should know the number of modes in advance, and if the number of modes is a lot, then too many generators should be designed. Thus, authors should verify the proposed network to be efficient in the limited number of k, even though the number of modes is much greater than k. I believe the mode collapse might not be reduced in this case. Please add the corresponding experiments to verify the efficiency of the proposed method.

Answer:

We supplemented an entire section, Section 4.4, to address your suggestion. In this section, we conducted a set of experiments on the synthetic dataset R15 to evaluate the influence of the hyperparameter  indicating the number of generators on the efficiency of AE-GAN. Based on the experimental results shown in Figure 8, we made a detailed discussion on the influence induced by different values of k, and provided a possible way of pursuing the approximate values of k.

Please refer to Section 4.4, on Pages 15-16.

  1. Regarding the overall cost function with the constraint for the encoder and the decoder scheme, it cannot be understood that regulating parameter lambda has a minus value. Please explain why the augmented cost function is added with minus lambda regulated constraint. Also, it needs to explain the value of lambda was selected as 0.5 in experiments.

Answer:

Exactly, the regulating parameter lambda should have a positive value. It was mistakenly marked with a minus in the writing process. The overall loss function is the cumulative sum of the three constituent losses. Lambda is used to maintain the balance of GAN loss and the Cross-entropy loss of clustering, preventing the two losses from losing efficacy in training. The value of lambda, 0.5, was selected from many trials in experiments. It can be viewed as a default from the perspective of experimental results.

We corrected all the involved formulas and contents. Please refer to Pages 7-9, and Algorithm 1 on Page 11.

  1. Regarding the figure1, this figure does not show the relation between ED scheme and discriminator. That is, overall cost function includes the constraint for ED to be optimized. However, the figure1 does not show any connection between these. Please correct the figure to show this relation in proper way.

Answer:

As suggested, we modified Figure 1 to explicitly illustrate the relation between the encoder-decoder and discriminator, and added some explanatory text accordingly. Please refer to Figure 1 on Page 5.

  1. I agree it can be reasonable approach that the ED scheme is added to match the mode and the corresponding generator by clustering the data and the sample outputs of generators and using each means. However, it is not clear why the constraint for ED is added to the original cost function of GAN scheme. This should be explained theoretically in detail.

Answer:

The objective function is composed of three constituents. The first constituent, the loss of GAN, is introduced to ensure that the generated data can fit the distribution of real data. The second is the cross-entropy loss of the generated data, which guarantees that the generated data of different clusters can cover different modes on one hand, and on the other hand, the generated data can be utilized as a reference to obtain the clustering labels of the real data, as it is supposed to be classified into the category of the nearest real data after clustering. Accordingly, the third is the cross-entropy loss of the real data, and its category originates from the nearest generated data. From the perspective of the  loss, it is expected that the model can distinguish both the real data in different modes and the generated data in different clusters, and when it converges, the generated data in different clusters can cover different modes of the real data and the labels of the generated data and real data are consistent.

We supplemented the above theoretical explanation as a separate paragraph to the manuscript. Please refer to Lines 273-284 on Page 7.

Round 2

Reviewer 2 Report

Comments and Suggestions for Authors

Most of comments for the previous version of the paper were satisfied by proper revision. However, the most important technical limitation regarding the number of generators, k was not analyzed properly. Since the proposed AE-GAN must work well only when k is matched to the number of modes, network needs the multiple generators as many as the number of modes. This is a crucial limitation in architecture level, especially for the large mode number cases.  Also, it means this network should be changed to the network with k generators scheme in every different mode number case to guarantee the accuracy. In the revised version of paper, even though authors experimented and analyzed the cases of different k from the number of modes in section 4.4, this is not enough. The additional experiments comparing with other models such as InfoGAN and MGAN in cases of small k are needed to show the relative efficiency of the proposed method even in the small k cases. Otherwise, this proposed algorithm is quite limited practically.

Author Response

Reviewer 2:

Most of comments for the previous version of the paper were satisfied by proper revision. However, the most important technical limitation regarding the number of generators, k was not analyzed properly. Since the proposed AE-GAN must work well only when k is matched to the number of modes, network needs the multiple generators as many as the number of modes. This is a crucial limitation in architecture level, especially for the large mode number cases.  Also, it means this network should be changed to the network with k generators scheme in every different mode number case to guarantee the accuracy. In the revised version of paper, even though authors experimented and analyzed the cases of different k from the number of modes in section 4.4, this is not enough. The additional experiments comparing with other models such as InfoGAN and MGAN in cases of small k are needed to show the relative efficiency of the proposed method even in the small k cases. Otherwise, this proposed algorithm is quite limited practically.

Answer:

First of all, we greatly appreciate the reviewer for the constructive comments.

We consider that the hyperparameter k of the proposed model AE-GAN might not just be limited to small numbers, and we shall explain this viewpoint in detail from the following two perspectives.

(1) From the perspective of architectures

First, according to the architectures listed in Appendix A, the number of parameters of each generator is approximately ranging from the magnitude of 104 to 105. Consequently, when k is set to numbers between 102 to 103, the overall number of parameters of AE-GAN is ranging from 106 to 108 accordingly. This upper bound approximates to the quantity of parameters of VGG 16, a generic deep neural network. Therefore, it seems feasible to set k to numbers that are not that small.

Second, if the network structure of the generators is given, the number of generators k may not influence the efficiency of model training heavily. The reason lies in that the time spent in training is used to generate samples and conduct subsequent operations on them, and the number of generated samples, i.e., the times of computation fulfilled by all generators, is totally dependent on the number of input samples, no matter how many generators are employed to participate this task. Of course, when k rises, the number of parameters increases, and more epochs of training can be needed accordingly. But the crucial factor that affects the efficiency of training is the quantity of samples of the used datasets.

Third, even though k is a parameter that controls the architecture of AE-GAN, it can be easily implemented by just inputting the value of parameter before training, provided that k is determined in advance.

(2) From the perspective of datasets

We conducted some statistics on synthetic and image datasets for the purpose of clustering, and the results are shown in Table 1 and Table 2, respectively. It can be seen that the number of classes in synthetic datasets ranges from 2 to 100, where the number of vast majority is less than 35, and the number of classes in image datasets ranges from 2 to 200, where the number of vast majority is no more than 20.

Clearly, the range of k for AE-GAN can cover all these datasets.

Most of the related models or algorithms are experimented on these datasets to validate their effectiveness and generalization for clustering. In the manuscript, we adopted 3 and 5 frequently used synthetic and image datasets, respectively. Their numbers of classes are 5, 15, 10, and 20, and their numbers of samples are 600, 788, 5000, 1440, 60000, and 70000. Likewise, we consider that the extensive experiments conducted on the datasets are sufficient to manifest the effectiveness and generalization of our model.

Table 1. Synthetic datasets for Clustering.

Number

Name of the Dataset

Classes

Number of Samples

1

Jain

2

373

2

Flame

2

240

3

Wdbc

2

569

4

Thyroid

2

215

5

Breast

2

699

6

Pathbased

3

300

7

Spiral

3

312

8

Wine

3

178

9

Iris

3

150

10

Aggregation-S

5

788

11

Asymmetric

5

1000

12

Overlap

6

1000

13

Skewed

6

1000

14

Glass

7

214

15

Unbalance

8

6500

16

Yeast

10

1484

17

R15

15

600

18

S1

15

5000

19

S2

15

5000

20

S3

15

5000

21

S4

15

5000

22

A1

20

3000

23

Letter

26

20000

24

D31

31

3100

25

A2

35

5250

26

A3

50

7500

27

leaves

100

1600

Table 2. Image datasets for Clustering.

Number

Name of the Dataset

Classes

Number of Samples

1

REUTERS

4

685071

2

REU-10k

4

10000

3

HHAR

6

10299

4

MNIST

10

70000

5

Fashion-MNIST

10

70000

6

CIFAR-10

10

60000

7

USPS

10

9298

8

STL-10

10

13000

9

imaginet-10

10

13000

10

imaginet-dog

15

19500

11

CIFAR-100

20

60000

12

Coil-20

20

1440

13

Tiny-imagenet

200

100000

Additionally, enlightened by the comments, we conducted some additional experiments on R15 to explore the possible range of the parameter k around the actual value in which AE-GAN can still perform effectively.

To this end, we revised Figure 8, inserted a paragraph to Section 4.4 and modified the paragraphs in its context accordingly. Please refer to Pages 15-17 in the revised manuscript.

Round 3

Reviewer 2 Report

Comments and Suggestions for Authors

The main issue regarding the number of generators, k was commented. Even though the comments are not fully satisfied, I suggest this paper to be published in this state. However, if possible, please add some analyses with the corresponding experimental results as follows.

1. Please add the comparison results of MGAN, infoGAN, and the proposed AE-GAN for the lower number of k than the actual mode number. I guess this experimental result can show the privilege from the additional Auto-Encoder scheme to reduce the mode collapse relatively more than MGAN, even though the result from the proposed scheme itself is also not good enough as in Fig. 8. In current revised version of the paper, there is only the comparison result and analysis for the k=15 case, which is same to the mode numbers. In this case, it is obvious that the proposed algorithm is better than others, because the proposed system is designed to be fit to the matched number of modes, k.

Author Response

Reviewer 2:

The main issue regarding the number of generators, k was commented. Even though the comments are not fully satisfied, I suggest this paper to be published in this state. However, if possible, please add some analyses with the corresponding experimental results as follows.

  1. Please add the comparison results of MGAN, InfoGAN, and the proposed AE-GAN for the lower number of k than the actual mode number. I guess this experimental result can show the privilege from the additional Auto-Encoder scheme to reduce the mode collapse relatively more than MGAN, even though the result from the proposed scheme itself is also not good enough as in Fig. 8. In current revised version of the paper, there is only the comparison result and analysis for the k=15 case, which is same to the mode numbers. In this case, it is obvious that the proposed algorithm is better than others, because the proposed system is designed to be fit to the matched number of modes, k.

Answer:

We are grateful for the reviewer’ valuable and helpful suggestion.

Following this suggestion, we modified the last three paragraphs of Section 4.4 and conducted a detailed comparison of the experimental results of AE-GAN with k = 14 and that of MGAN and InfoGAN to manifest the privilege of AE-GAN even with a value of k lower than the actual number of modes.

Please refer to Pages 16-17.
